# Support and follow-up needs of patients discharged from intensive care after severe COVID-19: a mixed-methods study of the views of UK general practitioners and intensive care staff during the pandemic's first wave

Ana Cristina Castro-Avila ![ORCID],[1,2] Laura Jefferson ![ORCID],[1] Veronica Dale ![ORCID],[1] Karen Bloor ![ORCID] [1]

[1]Department of Health Sciences, University of York, York, UK
[2]Carrera de Kinesiología, Universidad del Desarrollo Facultad de Medicina Clínica Alemana, Santiago, Chile

**Correspondence to**
Dr Ana Cristina Castro-Avila;
ana.castro@york.ac.uk

## ABSTRACT

**Objectives** To identify follow-up services planned for patients with COVID-19 discharged from intensive care unit (ICU) and to explore the views of ICU staff and general practitioners (GPs) regarding these patients' future needs and care coordination.

**Design** This is a sequential mixed-methods study using online surveys and semistructured interviews. Interview data were inductively coded and thematically analysed. Survey data were descriptively analysed.

**Setting** GP surgeries and acute National Health Service Trusts in the UK.

**Participants** GPs and clinicians leading care for patients discharged from ICU.

**Primary and secondary outcomes** Usual follow-up practice after ICU discharge, changes in follow-up during the pandemic, and GP awareness of follow-up and support needs of patients discharged from ICU.

**Results** We obtained 170 survey responses and conducted 23 interviews. Over 60% of GPs were unaware of the follow-up services generally provided by their local hospitals and whether or not these were functioning during the pandemic. Eighty per cent of ICUs reported some form of follow-up services, with 25% of these suspending provision during the peak of the pandemic and over half modifying their provision (usually to provide the service remotely). Common themes relating to barriers to provision of follow-up were funding complexities, remit and expertise, and communication between ICU and community services. Discharge documentation was described as poor and lacking key information. Both groups mentioned difficulties accessing services in the community and lack of clarity about who was responsible for referrals and follow-up.

**Conclusions** The pandemic has highlighted long-standing issues of continuity of care and complex funding streams for post-ICU follow-up care. The large cohort of ICU patients admitted due to COVID-19 highlights the need for improved follow-up services and communication between specialists and GPs, not only for patients with COVID-19, but for all those discharged from ICU.

## Strengths and limitations of this study

► This is the first study exploring National Health Service staff views of post-intensive care unit follow-up services to support patients recovering from severe COVID-19.

► Responses to our survey had good geographical spread but were limited in number and relied on volunteers.

► General practitioner interviews were small in number, but our findings align with those of larger studies conducted before the pandemic.

## INTRODUCTION

The significant physical, mental and cognitive problems patients face following a stay in an intensive care unit (ICU) are well documented.[1 2] Including prolonged muscle weakness, cognitive dysfunction and symptoms of post-traumatic stress disorder (PTSD), these are often collectively referred to as 'post intensive care syndrome' (PICS)[3] and may persist even 5 years after leaving hospital.[4–7] Six months postdischarge, 25% of survivors suffer severe disability[8] and only around 55% have returned to work.[9] Psychological disorders including depression, anxiety and PTSD are common, affecting 55% in the first year following ICU discharge.[10] The variety and severity of sequelae vary substantially.

The COVID-19 pandemic has resulted in a large and rapid increase in intensive care activity, which will challenge post-ICU services in several ways. Increases in ICU capacity necessitated the use of less experienced staff and very high workloads. More stringent infection control protocols created new potential causes of distress, including not

allowing family members inside the unit and healthcare professionals wearing personal protective equipment (PPE). These factors might lead to a very large cohort of critical illness survivors that might have greater than expected needs due to the context and length of their critical care stay,[11] which could put the current capacity of services under stress.

Post-ICU follow-up from hospital teams is likely to have been compromised during the pandemic. The availability, form and scale of services for ICU survivors of COVID-19 are largely unknown and patient needs are difficult to predict. Poor communication and blurred lines of responsibility between secondary and primary care were identified as potential barriers to post-ICU care continuity before the pandemic.[12 13] Timely, appropriate support could potentially prevent future problems in patients' physical, cognitive and mental health and care; identifying how and to what extent these services have been scaled up is important to inform the future response of the health service. This study aims to identify follow-up services that were available during and after the UK's first wave of the COVID-19 pandemic, early reflections on care during the first wave, and the views of critical care staff and general practitioners (GPs) about these patients' future needs and care coordination.

## METHODS

We employed sequential mixed methods following a pragmatic approach. We collected data through online surveys and semistructured telephone interviews with clinicians leading post-ICU follow-up and GPs.

In collaboration with clinicians in the field, we developed a very brief questionnaire of ICU staff to determine usual procedures of follow-up of patients after ICU discharge and changes during the pandemic. The Intensive Care Society, Faculty of Intensive Care Medicine, British Association of Critical Care Nurses and leading experts on intensive care disseminated this survey through newsletters, targeted emails and Twitter. We invited clinicians managing follow-up care of discharged ICU patients to participate and asked respondents to volunteer for interviews. We sampled volunteers purposively by geographical location and their responses to three of the survey questions: number of extra staffed beds opened during the pandemic, if they were offering follow-up services during the pandemic and whether the provision of follow-up had changed.

In collaboration with the Royal College of General Practitioners (RCGP), we developed and distributed a very brief questionnaire exploring GPs' awareness of post-ICU follow-up services and broad concerns about care of patients with severe COVID-19. The RCGP also included three of these questions in a routine survey of their GP research panel. GPs proved difficult to recruit to interviews through the survey, so we supplemented this with 'snowballing' using contacts at the RCGP, University of York and The King's Fund. We attempted to generate geographical spread in terms of location and COVID-19 incidence.

Both surveys were piloted with clinicians and experts. Initially, we shared the aim of the survey and the questions, so experts in the field could assess to what extent the questions provided the information needed, and language was precise, clear and concise. We then tested the survey with our target clinicians to ensure clarity and identify any essential omissions. We used Qualtrics XM software (Qualtrics, Provo, Utah) to distribute the surveys and collect responses.

Mindful of time pressures for these health professionals at this time, we limited interviews to 30 min and designed topic guides to answer key questions around follow-up provision, aiming to generate more indepth knowledge than could be achieved through survey alone. Interviews with ICU clinicians leading follow-up services explored views on whether and how the future needs of patients with COVID-19 differed from patients without COVID-19 and captured early reflections on ICU care and transitions during the first wave of the COVID-19 outbreak. Interviews with GPs explored their prior experience of managing post-ICU patients and information needs in relation to severe COVID-19. Potential participants were given participant information leaflets and consent forms, detailing the ethical considerations (available as online supplemental material), and verbal consent was obtained prior to each interview.

### Data analysis
#### Quantitative
Survey data were exported to IBM SPSS Statistics for Windows (V.26) for analysis. Absolute and relative frequencies were used to summarise responses. We calculated the average rank for the question about GPs' concerns regarding future care needs of patients recovering from a COVID-19-related critical care stay.

#### Qualitative
Interviews were audio-recorded and, as is increasingly being adopted in rapid qualitative research,[14] analysis was undertaken directly from audio recordings and detailed notes. Sections were transcribed for use as quotations. We assigned a code and a number to each audio recording and transcription to ensure anonymity.

An inductive approach was taken to analysing data using thematic analysis according to the six steps outlined by Braun and Clarke (p35)[15] (table 1). Following familiarisation with the data, two researchers (ACC-A and LJ) developed an initial coding framework of the main themes, which were discussed with the wider team and topic experts to refine the framework and distil overarching themes. Some representative quotes are presented to contextualise and aid interpretation.

A reflexive approach was taken during the analysis process to consider how researchers' presence and a priori assumptions may have influenced the data collection and analysis stages. The lead researcher (ACC-A) is a trained

**Table 1** Stages of qualitative thematic analysis based on Braun and Clarke[15]

| Phases of thematic analysis | Description |
|---|---|
| Familiarisation with the data | Interview notes were made immediately following each interview and were reviewed multiple times, along with audio recordings. |
| Generating initial codes | Initial codes were developed into a framework through discussion between the two primary coders (ACC-A and LJ) with wider team members and topic experts. Data, in the form of interview notes with transcribed sections, were sorted using data management software (NVivo V.12, QSR International). The same coding framework was applied across both participant groups: GPs and ICU. |
| Searching for themes | Initial codes were categorised into overarching themes, drawing out similarities and differences across participant groups. The 9 initial major themes and 21 subthemes were revised in line with the aims of this research. |
| Reviewing themes | Themes were reviewed and refined by the study team. Overlaps were reorganised into higher-order themes. For example, 'communication' appeared as a challenge for direct patient care, delivering news to family members and ensuring continuity of care outside of the hospital. Therefore, communication was reorganised into a higher-order theme comprising these three different aspects. |
| Defining and naming themes | A final framework of four themes and three subthemes was developed. |
| Producing the report | Qualitative and survey findings were integrated, combining the more descriptive qualitative findings with those from the survey that pertains to the ICU environment and provision. More 'thematic' qualitative findings were summarised separately. Participant quotes were used to illustrate key points and provide transparency in this process. |

GP, general practitioner; ICU, intensive care unit.

ICU physiotherapist, which influenced the research question chosen and could have affected the themes that are presented in this article. Through regular discussion among the research team as themes were developed and refined, we limited any impact of researcher bias on the process of data analysis.

### Patient and public involvement

We reviewed and discussed the project protocol with our patient and public involvement group, refining it in response. We were, however, unable to capture patients' views on follow-up services in our study timescale.

### Dissemination to participants and related patient and public communities

The findings of the wider programme of research were shared with all our participants to check our interpretations captured their views. All the participants who replied confirmed our interpretations were accurate.

### RESULTS
### Descriptive findings

Between 15 June and 3 August, ICU follow-up lead clinicians from 112 units (43% of acute National Health Service (NHS) Trusts in England) responded to our survey, of whom 83% were based in England and 96% were from mixed intensive care and high dependency units. On average, units more than doubled bed capacity at the height of the first wave (table 2).

Follow-up services were offered in 80 units (71% of those sampled); of these, 20 reported ceasing provision and 53

modifying provision of services during the pandemic. Eight units implemented a new follow-up service after the peak of the pandemic. Occupational therapy and physiotherapy were the services with the greatest increase (table 2).

Fifty-eight GPs responded to our survey and an additional 537 responded to three questions we distributed via the RCGP (table 3). Of the RCGP responses, 78% came from England, 61% were female, 63% were 35–54 years old and 83% were white. Over 60% of GPs were unaware of the follow-up services generally provided by their local hospitals and whether or not these were functioning during the pandemic. On average, four patients from their patients' list had been through ICU due to severe COVID-19. Physical and mental healthcare needs were ranked similarly high in terms of areas of concern with future patients recovering after a critical care stay.

We conducted 23 interviews between 23 June and 30 July: 17 with ICU staff (7 ICU consultants, 7 senior nurses, 3 rehabilitation coordinators) and 6 GPs. ICU interviews covered all UK regions, with the ICUs having an average capacity before the pandemic of 14 beds (range 4–60), increasing by 16 beds on average (range 2–38 beds). The GPs covered different regions of England and a mix of patient demographics.

Since the aim of our interviews was predominantly to provide a more in-depth descriptive account of current ICU provision, much of these findings are summarised descriptively under the sections 'ICU environment during the COVID-19 first wave' and 'Provision of follow-up

**Table 2** Responses of ICU leads about follow-up services during the pandemic

| Information about the unit | n=112 | Responses |
|---|---|---|
| When is the critical care discharge summary sent to the patient's general practitioner? | 38 | When the patient is discharged from hospital |
| | 36 | After critical care discharge, but before discharge from hospital |
| | 16 | Other |
| | 8 | I do not know |
| When is the first follow-up? | 29 | 2–3 months after discharge from hospital |
| | 28 | 2–3 months after discharge from critical care |
| | 11 | Other |
| | 8 | 4–6 months after discharge from hospital |
| | 3 | 1 month after discharge from hospital |
| Number of beds in your unit, mean (SD) | 93 | Before: 13.9 (11.1) |
| | | During peak: 33.7 (31.0) |
| | | Change: 20.1 (23.9) |

| Changes reported | n=53 | Details of change | | |
|---|---|---|---|---|
| Change in the format of the contacts (eg, remote consultations) | 39 | Remote consultations via telephone or video call | | |
| | 15 | Face-to-face clinics in hospital wearing personal protective equipment | | |
| | 2 | Home visits | | |
| Change in the number of professionals involved, mean (SD) | 22 | Before: 2.8 (1.9) | | |
| | | After: 4.1 (2.4) | | |
| | | Change: 1.3 (2.6) | | |
| Change in the timing of the first contact | 24 | Time before first follow-up contact is shorter than usual | | |
| | 2 | Time before first follow-up contact is longer than usual | | |
| Services available | 25 | | **Before** | **During** |
| | | Review of ICU history/diary and ICU events with patient | 23 | 22 |
| | | Assessment of sleep | 15 | 12 |
| | | Physiotherapy | 13 | 17 |
| | | Medicines reconciliation | 10 | 8 |
| | | Psychology | 9 | 10 |
| | | Assessment of sexual function | 8 | 4 |
| | | Dietetics | 6 | 5 |
| | | Speech and language therapy | 5 | 6 |
| | | Cognitive assessment | 5 | 4 |
| | | Psychiatry | 2 | 2 |
| | | Social work | 2 | 0 |
| | | Occupational therapy | 1 | 5 |

ICU, intensive care unit.

services'. More thematic findings are summarised under the themes 'barriers' and 'opportunity for change'.

## ICU environment during the COVID-19 first wave

All interviewees reported opening new areas and bringing nurses from other areas (eg, theatre, surgical recovery, other hospital wards) particularly those with ICU training. Consultants increased the frequency of their rotations to ensure continuous coverage. Administrative tasks for clinicians were suspended and all staff providing outpatient or outreach services returned to inpatient activities. ICU nurses split their time between patient care, staff supervision and training new staff, which was reported to increase workload and stress. Hospitals with greater bed capacity implemented proning and intubating teams, and some implemented retrieval teams to transfer patients between hospitals.

Numerous ICU interviewees mentioned that patients with COVID-19 may represent a new patient group, but

**Table 3** Responses from general practitioners survey

| Question | Responses | Our sample | RCGP |
|---|---|---|---|
| Does your nearest hospital Trust have specific follow-up services for all patients who have been discharged from critical care? n (%) | n | 58 | – |
| | I do not know | 36 (62) | – |
| | Yes | 11 (19) | – |
| | No | 11 (19) | – |
| Is the follow-up service functioning during the COVID-19 pandemic? n (%) | n | 45 | – |
| | I do not know | 39 (87) | – |
| | Yes | 6 (13) | – |
| Within your patient list, are you aware of any patients who have required critical care for severe COVID-19? n (%) | n | 56 | 537 |
| | Yes | 33 (59) | 208 (39) |
| | No | 13 (23) | 244 (45) |
| | I do not know | 10 (18) | 85 (16) |
| How many of your patients went through critical care due to severe COVID-19? | n | 24 | 462 |
| | Mean (min–max) | 4.4 (1–20) | 4.4 (0–50) |
| Considering future patients in your practice recovering from a COVID-19-related critical care stay, please rank your concerns about their care, mean rank (SD) | n | 40 | 447 |
| | Physical healthcare | 1.9 (1.4) | 1.4 (1.3) |
| | Mental healthcare | 2.4 (1.1) | 1.4 (1.2) |
| | Access to rehabilitation services | 3.1 (1.3) | 1.6 (1.3) |
| | Cognitive functioning | 3.5 (1.2) | 1.8 (1.3) |
| | Access to social care | 4.0 (1.1) | 1.7 (1.4) |

RCGP, Royal College of General Practitioners.

are still ICU survivors, with the weakness, mental and cognitive problems these patients commonly suffer. They expected patients with COVID-19 to suffer a longer-lasting deterioration of lung function, potential issues with renal function, a high incidence of shoulder injuries due to proning and cognitive problems related to the incidence of delirium.

Some thought it was too early to tell whether they will experience more physiological and psychological problems, but many highlighted particular treatments, including prolonged and deep sedation, opioids and neuromuscular blockers, which are associated with increased risk of muscular weakness, polyneuropathy and cognitive impairments. Patients experienced extended periods in a prone position, mechanical ventilation and less experienced nursing staff. One consultant believed that actively screening for mental health problems was needed (ICUcons09, Scotland).

One ICU nurse (ICUnurse04) who regularly administers mental health questionnaires to ICU patients had observed results from those with COVID-19. She reported that the ventilated patients had similar psychological issues as pre-COVID-19 ICU patients, but those who received continuous positive airways pressure (CPAP), and were therefore conscious, had worse scores. An ICU consultant echoed this and also highlighted potential difficulties due to PPE:

The other people in the bay watched [another patient] die over a number of days… It doesn't surprise me that the people here perhaps, more awake and aware are very, very traumatised by the experiences […] [Some patients] have delusional thoughts. I mean, I think that's gonna be a lot worse when you're surrounded by someone wearing a hazmat suit. (ICUcons06, South West)

### Provision of follow-up services

Before COVID-19, most ICU interviewees reported having a post-ICU follow-up service; the few who did not were planning to implement one after the pandemic. Most follow-up services were suspended during the peak of the first wave, as staff returned to in-hospital clinical duties. The few places that continued to provide such services used telephone follow-up, delivered by staff that were shielding.

Reported provision varies greatly, with some units delivering follow-up with just a consultant and/or a senior nurse, while others have multidisciplinary teams (MDTs). Some units start their follow-up during the ICU stay and have designated professionals to assess, refer and follow patients during the hospital episode and into the community. Others with well-established follow-up services refer ICU patients to pulmonary or cardiac rehabilitation services to recover fitness and muscle strength.

All unit staff we interviewed follow patients up 2–3 months after ICU discharge, but a minority also routinely call patients weekly (ICUnurse04, North East) or monthly (ICUnurse08, East Midlands). All had to change the format of their follow-up during the pandemic, and most replaced clinics with telephone calls or virtual consultations. One senior nurse highlighted the challenges of these virtual contacts due to reduced non-verbal communication and time limits:

> Phone calls don't really cut it because unless you're very skilled at talking to people, assessing people, you're not going to pick up on all those cues that people give out […] if we've got half an hour appointment, we won't get much out from in 10 minutes, but they'll open up. (ICUnurse14, East of England)

Two ICU interviewees said that they were implementing separate clinics for patients with COVID-19 to carry out extra recommended assessments, such as a chest X-ray at 6 weeks postdischarge as recommended by the British Thoracic Society.[16]

In some locations COVID-19 rehabilitation hospitals have been set up to provide specialist care and a "step down" for patients "that are not quite well enough to leave the acute setting and not quite well enough to go home" (GP1003, Yorkshire). This provided the opportunity for expert care to be delivered but relied on the Clinical Commissiong Group funding and "proactive planning for the worst-case scenario" (GP1003, Yorkshire).

GPs were concerned about the complex psychological needs of patients recovering from severe COVID-19 and that greater emphasis is placed on the physical needs of patients, with insufficient consideration of psychological support. All ICU interviewees agreed about the need for increased psychological support services.

Some ICU interviewees questioned the capacity of community rehabilitation services to improve patients' functioning. Both GPs and ICU staff felt that previous notions of thresholds for functional status post-ICU (several commented on assessments of patients climbing a flight of stairs) were arbitrary and not suitable for the wider age group of patients affected by COVID-19:

> Community Services work at getting someone functional. They don't work at getting them back to the state that they were at before they came into hospital. So, considering that a lot of our patients were younger patients, walking with a Zimmer frame to and from a bathroom aren't really what they want to be doing. They want to be getting back to their fitness level and back to work. (ICUrehab15, Wales)

One GP commented that: "it's only once they are home that the true level of need is understood" (GP1005, North West); at this point primary care and community services need to step in, but the support needs of these patients may be beyond their expertise and the capacity of services.

Some GP practices had taken proactive steps to follow up patients discharged from ICU (before and since COVID-19). One practice, pre-COVID-19, developed a list of 'at risk' patients who were monitored by a nurse practitioner and discussed at daily practice meetings, with the aim of reducing hospital admissions. One London practice had instigated weekly follow-up calls with patients with COVID-19 discharged from ICU, following a 'near miss' event whereby a serious complication had been detected opportunistically during a GP follow-up call.

### Thematic findings
#### Barriers to provision of follow-up services
Interviewees commented on various components that acted as barriers to the provision of follow-up services, relating predominantly to funding complexities, remit and expertise, and communication between ICU and community services.

##### Funding complexities
ICU interviewees in England felt the lack of a tariff for funding ICU follow-up clinics created variation in service provision. Both ICU and GP interviewees believed that community teams similar to those for stroke or cardiac rehabilitation should be set up for post-ICU patients.

Several interviewees were concerned that already overstretched community services with existing waiting lists could not meet the increase in demand from COVID-19 without improved funding and infrastructure. One GP described the closure of some community services, and ICU interviewees had concerns that those discharged from ICU did not have anywhere to go.

> We've had patients in tears, we've had seven patients through telephone calls. And all of them are absolutely distraught and feel like they've been abandoned in the community […] because they were thrown out of hospital very quickly. There's no services in the community for them at all. (ICUrehab15, Wales)

##### Remit and expertise
Both ICU staff and GPs found referrals to community follow-up services difficult, with differences in opinion about whose responsibility this was, as well as problems with waiting lists (particularly for mental health services). Community rehabilitation services were described as "patchy" (ICUnurse08, East Midlands). Both staff groups felt that hospital services were better placed to follow up patients discharged from ICU because they have a better understanding of the patient's needs. There appears, however, to be a general lack of awareness about the difficulties of coordinating patients' needs in each setting.

> I'm not sure the hospitals are always very aware of what services are available in the community… To give you a COVID example, I had a doctor ring me up and say 'he's been in hospital for a long time, can you make sure he sees a psychiatrist when he comes out?'… 'no, I don't have that kind of access to psychiatrists'. (GP1001, East)

### Communication

Poor or delayed communication can result in misunderstandings about patients' support needs, and GP interviews highlighted that these high-risk patients could potentially suffer adverse events if their follow-up is not adequate. Hospitals and GPs communicate through discharge letters, which all GPs described as inadequate, often produced by junior doctors and lacking pertinent detail: "the nuance, the detail is often missing" (GP1005, North West).

Interviewees also reported examples of good practice, for example respiratory consultants sharing contact details and working closely with GPs when patients with severe COVID-19 are discharged. In some cases, ICU interviewees commented on the importance of long-standing professional relationships with community rehabilitation service providers.

GPs were concerned about the evolving nature of COVID-19 and changing medical understanding. They welcomed specific and targeted information that would help them to guide patients' care after an intensive care and hospital stay. Others suggested a need for better communication with hospital teams to develop their understanding of specific patients' needs and where to find support. One GP summarised the information needs as "what to look for, when to refer back into hospital and types of patients that need specific follow-up" (GP1002, Yorkshire).

All GPs stressed that guidance needs be balanced and channelled through a respected national body, as they faced an overload of information, described by one as "guideline fatigue" (GP1004, London). This was particularly cumbersome during the early phases of the pandemic when sometimes conflicting information was disseminated daily, from multiple sources.

To cope with the levels of information during the first wave, some GP practices had initiated daily team meetings to discuss and keep abreast of key changes. One GP commented that the vast amount of COVID-19 information hampered GPs from employing their "generalist skills" to tailor care to the individual's needs (GP1005, North West).

### The pandemic as an opportunity to change

Interviewees from three ICUs described the pandemic as an opportunity to initiate an MDT follow-up clinic by making visible the issues faced by patients discharged from ICU.

> [We've had an] uplift in the therapy staff […] [as] we've now got more dietician[s], more physio, more pharmacy, we've never had an OT before six months ago, we've never had a psychologist of our own […] We're now in a position to offer MDT follow-up service rather than just a simple follow-up clinic. (ICUcons06, South West)

Interviewees highlighted the need for increased provision in response to the pandemic, resulting from large numbers of newly affected patients, uncertainties in their support needs and a younger population needing to return to work.

## DISCUSSION

### Statement of principal findings

The peak of the first wave of COVID-19 saw dramatic changes in ICUs to increase bed capacity. This was accompanied by adaptations to (and, in general, reductions in) the follow-up care provided, although most units retained some form of follow-up service.

Before COVID-19, there was a perception that funding streams and referral systems may hinder provision. The lack of a tariff for post-ICU follow-up may cause unwarranted variations, which interviewees believed could be addressed through a 'reablement after critical care pathway' similar to that in place for cardiac and stroke rehabilitation.

Again, before the pandemic, communication between primary and secondary care was sometimes poor, and care was hampered by a lack of clarity about responsibilities for meeting various post-ICU patients' needs. GPs expressed a need for specific information about recovery from critical illness, collated by a single, authoritative professional group.

All of these existing constraints were believed to have been magnified by the COVID-19 pandemic.

### Strengths and limitations of the study

To the best of our knowledge, this is the first study exploring NHS staff views on follow-up services post-ICU and plans to support patients recovering from severe COVID-19.

Our recruitment strategy relied heavily on social media due to time constraints, which might have attracted participants who are more willing to share their opinions. Potentially, a different recruitment approach might have yielded different results. While we cannot guarantee that our survey samples are representative of the UK, responses were spread across the country, covering different ICU unit sizes, increases in capacity and sizes of NHS Trusts, and are similar to those reported by Connolly *et al*,[17] who used a different recruitment strategy and included a larger sample. GP responses to the survey were low, but they were spread geographically and had similar views to the larger sample from the RCGP survey. GPs tend to see very few patients who have been discharged from intensive care,[12] and it is not clear how this might affect our results considering that participants were self-selected.

While our qualitative interviews did not seek to achieve generalisability, our GP interview findings were consistent with each other and similar to those with larger samples conducted before the pandemic.[12 13] This study formed part of a larger project and was conducted rapidly to inform UK health policy during the peak of the pandemic. Nevertheless, our qualitative methods followed principles considered to promote rigour in qualitative research. Investigator and methodological

triangulation methods were employed to develop rich and indepth understanding; findings are evidenced using quotations to enhance the transparency and trustworthiness of conclusions drawn; findings were shared with our interviewees to ensure that our interpretations were accurate; and a reflexive approach was adopted to consider how researchers' a priori assumptions may have affected data collection and analysis.

## Meaning of the study: possible explanations and implications for clinicians and policymakers

A number of issues raised in this study are long-standing and have been highlighted in previous research[12 13]: inadequate discharge summaries, lack of clarity of responsibility for postacute patient care, fragmented and delayed communication, and limited knowledge regarding the support needs of post-ICU patients. During the pandemic, there has been RCGP training about the main post-ICU sequelae and potential treatments,[18 19] which could help to improve awareness around the mental, physical and cognitive consequences of an ICU stay. Problems in continuity of care, however, may need a joint approach to improve local organisation of care and how information is delivered across settings considering a 'whole patient journey'.[13] Discharge summaries written by more senior staff in hospital highlighting potential red flags and greatest awareness in secondary care regarding the capabilities of primary care were suggested as elements that could improve the communication and transition between secondary and primary care.

Commissioning and funding streams seem to be a major issue, as follow-up is recommended but not directly funded, unlike the pathways for cardiac and stroke rehabilitation, which were suggested by interviewees as models for post-ICU care. The evidence base for post-ICU follow-up is however partial and would benefit from further research.[20 21] Interviewees highlighted that the specialised nature of post-ICU care meant that intensive care staff were better placed to understand and refer patient to services for cognitive, physical and mental health problems, but funding did not always allow this. Additionally, interviewees suggested that services that were required for longer such as talking therapies and physical rehabilitation should be delivered in the community, where they might be more easily accessed.

Community rehabilitation services were described as 'patchy', with long waiting lists, an issue recognised by NHS England.[22] Recent initiatives to improve provision were welcomed, but some interviewees questioned whether the criteria for determining community rehabilitation needs were fit for purpose for younger, fitter populations and whether community rehabilitation services could change provision without extra funding to enhance infrastructure. Community mental health services were particularly recognised as overstretched with long waiting lists that prioritise patients at high risk of harming themselves or others.[23] Murray *et al*[24] suggest a model such as the Nightingale Hospital, but for rehabilitation, during and after the pandemic.

Our interviewees suggested that most of the long-term consequences faced by patients with COVID-19 are similar to those faced by others experiencing ICU. Knowledge about the sequelae of COVID-19 is at an early stage, and research on longer-term consequences of COVID-19, such as the PHOSP-COVID (Post-hospitalisation COVID-19) study following more than 10 000 patients for more than 12 months, will shed light on which sequelae relate to being critically ill more generally and which are specific to COVID-19. This should complement what is already known about PICS and effective treatment models.[25]

One interviewee mentioned that patients who received CPAP reported worse mental health than patients ventilated invasively. While this was only reported by one interviewee and should therefore be interpreted with caution, it may warrant further exploration in wider samples or research as, according to the Intensive Care National Audit and Research Centre report on 9 October 2020, 44% of patients with COVID-19 in critical care settings were not mechanically ventilated during the first 24 hours.[26] This implies that a high proportion of patients are awake and aware of their surroundings. Depending on the criteria for prioritisation, they may not qualify for long-term follow-up, and consequently might suffer from mental health symptoms without receiving formal support. Given the widespread management of patients with COVID-19 with CPAP and high-flow nasal cannulas, this cohort may need at least as much follow-up as those ventilated more invasively.

## Unanswered questions and future research

Follow-up services vary greatly, but the extent to which variations in provision are linked to differences in long-term outcomes is not clear. Identifying models of care which yield the best outcomes in the most efficient way could help develop the evidence base for reducing unwarranted variations in the future. The potential effect on mental health of being in intensive care while receiving CPAP may merit further research.

Patients who have had an ICU stay might show impairments even 5 years after discharge. Currently, appropriate length of follow-up is unclear, as is the point at which care should be continued in primary and community care settings only. Current National Institute for Health and Care Excellence guidance[27] addresses the early stage of follow-up, but not longer-term support.

The large cohort of younger than average ICU patients provides an opportunity to assess these services and ensure they meet the needs of those recovering from COVID-19 and other future patients discharged from intensive care.

**Acknowledgements** We would like to acknowledge the contribution and advice of the Faculty of Intensive Care Medicine, Intensive Care Society, Royal College of General Practitioners, Dr Philip Antill, Dr Shanthi Antill, Professor Yvonne Birks, Dr Bronwen Connolly, Dr Tom Lawton, Professor Hugh Montgomery, Professor Rupert

Pearse, Professor Zudin Puthucheary, Professor Trevor Sheldon and Professor Najma Siddiqi.

**Contributors** This study was designed and conceived by ACC-A, LJ and KB. VD conducted the survey data analysis. ACC-A and LJ conducted the interviews and their analysis. ACC-A wrote the first draft of this manuscript. LJ and KB made comments on all the versions of this manuscript. All authors have read and agreed on the final version.

**Funding** This article is an independent research commissioned and funded by the NIHR Policy Review Programme through the Partnership for Responsive Policy Analysis and Research (PREPARE) (grant number NIHR 200702). The research was carried out between 15 April and 11 August. These findings were presented to key stakeholders at the Department of Health and Social Care. The funders had no role in considering the study design or in the collection, analysis and interpretation of data, writing of the report, or decision to submit the article for publication.

**Disclaimer** The views expressed in this article are those of the authors and not necessarily those of the NIHR or the Department of Health and Social Care.

**Competing interests** None declared.

**Patient consent for publication** Not required.

**Ethics approval** This project was reviewed and approved by the Department of Health Sciences Research Governance and Ethics Committee at the University of York (ID number HSRGC/2020/397/A).

**Provenance and peer review** Not commissioned; externally peer reviewed.

**Data availability statement** Data from the interviews and survey responses are available upon reasonable request.

**ORCID iDs**
Ana Cristina Castro-Avila http://orcid.org/0000-0003-4475-4325
Laura Jefferson http://orcid.org/0000-0003-2139-3555
Veronica Dale http://orcid.org/0000-0003-3783-2030
Karen Bloor http://orcid.org/0000-0003-4852-9854

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
