## [Reviewer comments · BMJ Open]

ARTICLE DETAILS

TITLE (PROVISIONAL)	Support and follow-up needs of patients discharged from Intensive Care after severe COVID-19: a mixed-methods study of the views of UK general practitioners and intensive care staff during the pandemic's first wave
AUTHORS	Castro-Avila, Ana; Jefferson, Laura; Dale, Veronica; Bloor, Karen

VERSION 1 – REVIEW

REVIEWER	Vollam, Sarah University of Oxford, Nuffield Department of Neurosciences
REVIEW RETURNED	22-Jan-2021

GENERAL COMMENTS	Thank you for the opportunity to review this manuscript. This is a very important area of research and use of mixed methods is novel and appropriate. However, I have some concerns with the manuscript as it stands, which I have outlined below. The key issue lies with the qualitative part of this study. Although you state you conducted a thematic analysis this is not reflected in the presentation of results, which is very descriptive, not presented in a thematic format and does not demonstrate synthesis of the data. In addition, there are significant gaps in reporting related to the qualitative methods used, including ethical considerations, and factors related to trustworthiness, credibility, transferability, etc. There was no reporting checklist with the manuscript and the use of a qualitative checklist may guide inclusion of these aspects. It is of concern that the only limitation discussed is that of generalisability, when this is not the aim of qualitative research. The section of limitations should be expanded to include some significant limitations of this study, understandable given the circumstances but still requiring acknowledgment, such as the response rate of the survey and combination of two methods of obtaining survey data. How can you ensure the same participants did not respond twice to the two different surveys? The survey results are really interesting but are lost in the paper. A paragraph summarising these in the results would help to emphasise this key information. The paper seems rather imbalanced towards the qualitative data at present. The discussion of findings could be supported by wider referencing.
--

	Small points: In the discussion what is meant by (Bruce J, personal communication)? Is this a pseudonym for a participant (pseudonyms are not used in the results section) and what is meant by personal communication? The abstract doesn't clearly state the methods and analysis used. The response rate for GPs could be stated.
--	--

REVIEWER	Pattison, Natalie University of Hertfordshire, School of Health and Social Work
REVIEW RETURNED	12-Feb-2021

GENERAL COMMENTS	Thank you for the opportunity to review this interesting article regarding support and follow-up needs of patients discharged from intensive care following severe Covid. Overall this is a valuable, well written paper that will contribute to the wider knowledge base of critical care recovery services. Background Introductory paragraph sets the scene nicely and refers to appropriate literature, providing a rationale for the part of the study aims. The GP-ICU interface literature is less well described (for instance Zilahi and O'Connor has been omitted and Bench et al's work - which is mentioned but only in the discussion), adding these in would add to the rationale for the need for the study. Clearly, there are significant deficits in the interactions/interface (and knowledge base of GPs) GPs and ICUs. Methods The survey was designed with clinicians in the field; could the authors please add in information about how was content of face validity and external validity determined? The sampling could be more clearly described in the methods. The invitation process for inviting clinicians leading follow-up care was not via the NHS, which means a fair proportion could have been missed. ICU leads (clinical directors of ICU) are not the same as clinicians leading follow-up care. The potential professional population being surveyed is very large (many thousand for both critical care and GPs). The innovative solution via embedding three questions in a RCGP questionnaire is to be commended as other work in this area has shown it is not easy to engage GP communities. Sampling frame: the authors need to expand on what is meant by 'need for expanded bed capacity' as determined how, and when? This was a rapidly changing situation during the first wave of the pandemic. It would be helpful to have more details on this where was this information drawn from please and what does ' changes implemented', please be more precise in describing this as a sampling framework. How did you ensure that the GPs had been sufficiently exposed to the phenomenon of caring for critically ill discharged patients? Other work has suggested GPs can see less than five people like this each year.
--

	Although I appreciate as a national survey and the fact that this was circulated through networks means that ethical review was not required it would be important nonetheless to describe ethical processes in the methods, such as anonymity. What was the rationale for approaching ICU leads, rather than leads of critical care follow-up services as this will have yielded very different interview data; medical/nursing ICU leads may not be aware of the full scope of ICU recovery services in their institution, and certainly not during the rapidly changing situation of the pandemic. What did you do to ensure only one response per organisation? Results/Discussion 43% response rate is reasonable for a survey of this type for ICU clinicians. The low GP response rate to the initial survey might also be related to how this was circulated (any reflections in the discussion on this?). P7 second paragraph: I would argue is not relevant to the research question so I would omit this data – it is also more fully explored in other studies of staff experience. P8 I would be cautious about using this secondary data in the first paragraph and would perhaps consider rewording to make it clear this reflects her experience and is not primary data. Was there any more data that pointed to deficits in the interface between acute (ICU or post-ICU) and primary care? This would be really valuable more novel and would be good to have more GP data in this section. P9 for instance highlights the need for community teams when other literature points to the need for specialist and ICU-led provision of care (some of which might be community-based, but underpinned by ICU expertise of the complex pathologies/social/emotional sequelae induced from critical illness). Three times in the results there are reports of worse outcomes with NIV – but this is secondary reported qualitative data and I think there needs to be more caution in how this qualitative data is presented (It could be this stems from those interviewee’s interpretations/perceptions). While this is highly plausible, a caveat to this effect would be helpful, particularly as this is used to frame future research – a more circumspect statement would be better. I’m not sure why ‘personal communication’ is used on p12 in the discussion? The points around variation in provision are well made. I think more could be drawn out in the discussion re: GP/ICU interface and how this could be better addressed, based on the findings and the literature (see earlier suggestions).
--	--

VERSION 1 – AUTHOR RESPONSE

Reviewer 1: Dr. Sarah Vollam, University of Oxford

The key issue lies with the qualitative part of this study. Although you state you conducted a thematic analysis this is not reflected in the presentation of results, which is very descriptive, not presented in a thematic format and does not demonstrate synthesis of the data.	Thank you for this observation. This partly arose due to limits on word count as we attempted to describe quite a large body of work within this paper. Additionally, our primary focus for this paper was to provide a descriptive account of the follow up services available for patients with severe COVID-19, hence the descriptive nature of some of our qualitative findings which attempted to provide more information than the survey alone could offer, but were necessarily descriptive in nature. We have taken these comments on board and attempted to separate the more thematic findings from the descriptive. In the first, descriptive, section, we now present contextual information regarding the ICU environment during the first wave and the provision of follow-up services. The second, 'thematic' section of the results addresses the findings that relate to barriers for the provision of follow up services and the pandemic as a potential opportunity to change. We considered that information about how the situation was different was relevant for understanding how the needs of these patients might differ once they reach the community.
In addition, there are significant gaps in reporting related to the qualitative methods used, including ethical considerations, and factors related to trustworthiness, credibility, transferability, etc. There was no reporting checklist with the manuscript and the use of a qualitative checklist may guide inclusion of these aspects.	We have included the SRQR checklist to make sure all relevant aspects of the methods were reported correctly. We have added more information with regard to qualitative methods, including the ethical aspects of this research. The participant information leaflets, and consent forms are included as part of the supplementary material. We added the following paragraph in the methods section:
	Mindful of time pressures for these health professionals at this time, we limited interviews to 30 minutes and designed topic guides to answer key questions around follow-up provision; aiming to generate more in-depth knowledge than could be achieved through survey alone. Interviews with ICU clinicians leading follow-up services explored views on whether and how the future needs of COVID-19 patients differed from non-COVID patients and captured early reflections on ICU care and transitions during the first wave of the COVID-19 outbreak. Interviews with GPs explored their prior experience of managing post-ICU patients, and information needs in relation to severe COVID-19. Potential participants were given Participant Information Leaflets and Consent Forms, detailing the ethical considerations (available as supplementary material) and verbal consent was obtained prior to each interview. The study was reviewed and approved by the University of York Department of Health Sciences Research Governance and Ethics Committee (ID Number HSRGC/2020/397/A). In reference to the qualitative data analysis, we have added to and edited the following section, with the addition of a table which

	describes the steps taken using the approach described by Braun and Clarke (2006): Interviews were audio-recorded, though, as is increasingly being adopted in rapid qualitative research;¹⁴ analysis was undertaken directly from audio-recordings and detailed notes. Sections were transcribed for use as quotations. An inductive approach was taken to analysing data using thematic analysis according to the six steps outlined by Braun and Clarke (2006, p35) and given in Table 1. Following familiarisation with the data, two researchers (ACA and LJ) developed an initial coding framework of main themes, which were discussed with the wider team and topic experts to refine the framework and distil overarching themes. Some representative quotes are presented to contextualise and aid interpretation. A reflexive approach was taken during the analysis process to consider how the researchers' presence and a priori assumptions may have influenced the data collection and analysis stages. The lead researcher (ACA) is a trained ICU physiotherapist, which influenced the research question chosen and could have affected the themes that are presented in this article. However, through regular discussion amongst the research team as themes were developed and refined, we were able to limit the impact of individual researcher biases on the process of data analysis.
It is of concern that the only limitation discussed is that of generalisability, when this is not the aim of qualitative research.	In the case of the GP survey, the responses that were collected with Qualtrics have the associated IP address and location of the respondent, so we can exclude cases that responded twice from the same location.
The section of limitations should be expanded to include some significant limitations of this study, understandable given the circumstances but still requiring acknowledgment, such as the response rate of the survey and combination of two methods of obtaining survey data. How can you ensure the same participants did not respond twice to the two different surveys?	Regarding the responses collected through the RCGP, we cannot guarantee that these come from different participants. We have expanded our discussion around limitations of the study, and agree with the reviewer that the purpose of qualitative research is certainly not to seek generalisability. When referring to generalisability we were referring to the survey findings, but it is useful to note that this was not clear and we have revised this section now. This now reads: Our recruitment strategy relied heavily on social media due to time constraints, which might have attracted participants that are more willing to share their opinions. Potentially, a more targeted approach to include those that are less willing to volunteer might have yielded different results. While we cannot guarantee that our survey samples are representative of the UK, responses were spread across the country, covering different ICU unit sizes, increases in capacity and sizes of NHS Trusts, and are similar to those reported by Connolly, et al ¹⁷ who used a different recruitment strategy and included a larger sample. GP responses to the survey were low but they were spread geographically, and had similar views to the larger sample from the RCGP survey. GPs tend to see very few patients that have been discharged from intensive care,¹² and it is not clear how this might affect our results considering that participants were self-selected. While our qualitative interviews did not seek to achieve generalisability, our GP interview findings were consistent with each other, and similar to those with larger samples conducted before the pandemic.^{12 13} This study formed part of a larger project and was conducted rapidly to inform UK health policy during the peak of the

	pandemic. Nevertheless, our qualitative methods followed principles considered to promote rigour in qualitative research. Investigator and methodological triangulation were employed to develop richer and more in-depth understanding; findings are evidenced using quotations to enhance the transparency and trustworthiness of conclusions drawn; findings were shared with our interviewees to ensure our interpretations were accurate; and a reflexive approach was adopted to consider how the researchers' a priori assumptions may have effected data collection and analysis.
--	---

The survey results are really interesting but are lost in the paper. A paragraph summarising these in the results would help to emphasise this key information. The paper seems rather imbalanced towards the qualitative data at present.	We have added a brief description of the general results of the GP survey: Over 60% of GPs were unaware of the follow-up services generally provided by their local hospitals, and whether or not these were functioning during the pandemic. On average, 4 patients from their patient's list had been through ICU due to severe COVID-19. Physical and mental health care needs were ranked similarly high in terms of areas of concern with future patients recovering after a critical care stay.
The discussion of findings could be supported by wider referencing	We have added contextual information and further references as suggested.
In the discussion what is meant by (Bruce J, personal communication)? Is this a pseudonym for a participant (pseudonyms are not used in the results section) and what is meant by personal communication?	We have erased this because it was creating confusion.
The abstract doesn't clearly state the methods and analysis used.	This has been added under the subheading "design"
The response rate for GPs could be stated.	Because of our varied efforts to recruit GPs for the survey, it is not possible to calculate a response rate because it is not clear what the denominator will be. The survey was distributed by RCGP, by twitter

	and by snowballing, so we have no clear evidence about how many GPs saw the survey request.
Reviewer 2: Prof. Natalie Pattison, University of Hertfordshire	
Background Introductory paragraph sets the scene nicely and refers to appropriate literature, providing a rationale for the part of the study aims. The GP-ICU interface literature is less well described (for instance Zilahi and O'Connor has been omitted and Bench et al's work - which is mentioned but only in the	Thank you – indeed, there are significant deficits in the interactions/interface (and knowledge base of GPs) GPs and ICUs which is helpful to frame the rationale. We have amended the paragraph as follows: In the case of primary care, poor communication and blurred lines of responsibility between the hospital and primary care had already been identified as potential barriers for care continuity post-ICU before the pandemic. Timely, appropriate support could potentially prevent future problems in patients' physical, cognitive and mental health and care; therefore, identifying how and to what extent these services have been scaled up is pivotal for the future response of the health service.
discussion), adding these in would add to the rationale for the need for the study.	The references to Zilahi and O'Connor (2019) and Bench et al (2016) have been added.
Methods The survey was designed with clinicians in the field; could the authors please add in information about how was content of face validity and external validity determined?	We have added this paragraph to the methods section: Both surveys were piloted with clinicians and experts. Initially, we shared the aim of the survey and the questions, so experts in the field could assess to what extent the questions provided the information needed, and language was precise, clear and concise. We then tested the survey with our target clinicians to ensure clarity and identify any essential omissions. In terms of the external validity of the findings, we amended the text of the discussion: While we cannot guarantee that our survey samples are representative of the UK, responses were spread across the country, covering different ICU unit sizes, increases in capacity and sizes of NHS Trusts, and are similar to those reported by Connolly, et al. GP responses to the survey and interviews were low but they were spread geographically, and all agreed on the challenges of organising care for patients discharged after an ICU stay.
The sampling could be more clearly described in the methods. The invitation process for inviting clinicians leading follow-up care was not via the NHS, which means a fair proportion could have been missed. ICU leads (clinical directors of ICU)	We have reworded how we refer to ICU staff. The invitation called for ICU staff in charge of post-ICU followup, which can be confirmed by our survey responses with a wide range of professions from ICU consultant, advanced critical care practitioner, cardiorespiratory team lead to sister nurses, follow-up lead and physiotherapists.

are not the same as clinicians leading follow-up care.	
The potential professional population being surveyed is very large (many thousand for both critical care and GPs). The innovative solution via embedding three questions in a RCGP questionnaire is to be commended as other work in this	Thanks for this comment. We are well aware of the difficulties of reaching GPs for research purposes, and I agree that it is a challenge, in particular, when there are time constraints.

area has shown it is not easy to engage GP communities.	
Sampling frame: the authors need to expand on what is meant by 'need for expanded bed capacity' as determined how, and when? This was a rapidly changing situation during the first wave of the pandemic. It would be helpful to have more details on this where was this information drawn from please and what does 'changes implemented', please be more precise in describing this as a sampling framework.	We used the answers to three of the survey questions to select our sample of ICU interviewees:  1. During the peak of the COVID-19 pandemic, how many beds were there in your unit (please estimate a number or range)? 2. Are you offering recovery or follow up services during the pandemic? 3. Are these recovery or follow up services different from pre-COVID-19 services? We have changed the wording of this section to make it clearer: We sampled volunteers purposively by geographical location, and their responses to three of the survey questions: number of extra staffed beds opened during the pandemic, if they were offering follow-up services during the pandemic, and whether the provision of follow-up was different.
How did you ensure that the GPs had been sufficiently exposed to the phenomenon of caring for critically ill discharged patients? Other work has suggested GPs can see less than five people like this each year.	We agree with the point made by the reviewer that GPs tend to not see many patients after an ICU stay. We have added this as a potential limitation related to recruiting participants on a voluntary basis. GPs tend to see very few patients that have been discharged from intensive care, and it is not clear how this might affect our results considering that participants were self-selected.

Although I appreciate as a national survey and the fact that this was circulated through networks means that ethical review was not required it would be important nonetheless to describe ethical processes in the methods, such as anonymity.	This study went through an ethics committee review, via the University of York not the NHS. The surveys had a brief consent form at the beginning. The interview participants received a consent form and a participant information leaflet, which are available as a supplementary file. We have added the following sentence in the data analysis: We assigned a code and a number to each audio-recording and transcription to ensure anonymity.
What was the rationale for approaching ICU leads, rather than leads of critical care follow-up services as this will have	We have reworded how we refer to the ICU staff we approached. We refer to ICU clinicians leading follow-up services. We had used ICU leads to make it shorter, but clearly, this is confusing.
yielded very different interview data; medical/nursing ICU leads may not be aware of the full scope of ICU recovery services in their institution, and certainly not during the rapidly changing situation of the pandemic.	
What did you do to ensure only one response per organisation?	At the beginning of the survey, participants provided the name of their NHS trust and hospital. We merged this information with a list of Acute NHS Trusts and all the hospitals in each of them to ensure only one response per hospital was collected. In the case of more than one response, we analysed the first response.
Results/Discussion 43% response rate is reasonable for a survey of this type for ICU clinicians. The low GP response rate to the initial survey might also be related to how this was circulated (any reflections in the discussion on this?).	We have included the following: Our recruitment strategy relied heavily on social media due to time constraints, which might have attracted participants that are more willing to share their opinions. Potentially, a more targeted approach to include those that are less willing to volunteer might have yielded different results.
P7 second paragraph: I would argue is not relevant to the research question so I would omit this data – it is also more fully explored in other studies of staff experience.	We have erased this.

P8 I would be cautious about using this secondary data in the first paragraph and would perhaps consider rewording to make it clear this reflects her experience and is not primary data.	We have reworded that paragraph to make clear that the observation made by the interviewee is based on data she regularly collects using standardised mental health questionnaires. Now it says: One ICU nurse that regularly administers mental health questionnaires to ICU patients, had the results from the cohort with COVID-19 reporting that ventilated patients had the same psychological issues as other ICU patients, but those who received continuous positive airways pressure (CPAP) and were therefore conscious, had worse scores.
--	---

Was there any more data that pointed to deficits in the interface between acute (ICU or post-ICU) and primary care? This would be really valuable more novel and would be good to have more GP data in this section. P9 for instance highlights the need for community teams when other literature points to the need for specialist and ICU-led provision of care (some of which might be community-based, but underpinned by ICU expertise of the complex pathologies/social/emotional sequelae induced from critical illness).	These two views are not necessarily mutually exclusive. It was recognised by ICU staff and GPs that those with more knowledge about critical care were better placed to conduct the initial follow-up to identify potential issues and refer to services. However, funding streams did not always allow them to follow that route. Also, therapy for mental or physical health issues requires several sessions, therefore, it was thought that those services should be provided in the community where it was easier to access them. We have added the following paragraph, which we hope explains this situation. Interviewees highlighted that the specialised nature of post-ICU care meant that intensive care staff were better placed to understand and refer patient to services for cognitive, physical and mental health problems, but funding did not always allow this. Additionally, interviewees suggested services that were required for longer such as talking therapies and physical rehabilitation should be delivered in the community, where they might be more easily accessed.
--	---

Three times in the results there are reports of worse outcomes with NIV – but this is secondary reported qualitative data and I think there needs to be more caution in how this qualitative data is presented (It could be this stems from those interviewee’s interpretations/perceptions). While this is highly plausible, a caveat to this effect would be helpful, particularly as this is used to frame future research – a more circumspect statement would be better.	We agree that the thoughts of one interviewee need to be interpreted with caution, but nevertheless this is primary research data that may merit further exploration. We have added a caveat to this statement in our discussion section: One interviewee mentioned that patients who received CPAP reported worse mental health than patients ventilated invasively. While this was only reported by one interviewee and should therefore be interpreted with caution, it may warrant further exploration in wider samples or research as, according to the Intensive Care National Audit and Research Centre (ICNARC) report to the 9th of October, 44% of COVID-19 patients in critical care settings were not mechanically ventilated during the first 24 hours.²⁶ This implies there is a high proportion of patients who are awake and aware of their surroundings, which depending on the criteria for prioritisation, may not qualify for long-term follow-up, and consequently, might suffer from mental health symptoms without receiving formal support. And for future research, it says: The potential effect on mental health of being in intensive care while receiving CPAP might merit further research given this is a group of patients that is not normally followed-up.
--	---

I'm not sure why 'personal communication' is used on p12 in the discussion?	It has been erased.
The points around variation in provision are well made. I think more could be drawn out in the discussion re: GP/ICU interface and how this could be better addressed, based on the findings and the literature (see earlier suggestions).	We have expanded on the issues in several parts of the discussion. For example: A number of issues raised in this study are long standing and have been highlighted in previous research: inadequate discharge summaries, lack of clarity of responsibility for post-acute patient care, fragmented and delayed communication and limited knowledge regarding the support needs of post-ICU patients. During the pandemic, there has been RCGP training about the main post-ICU sequelae, and potential treatments, which could help improve awareness around the mental, physical, and cognitive consequences of an ICU stay. Problems in continuity of care, however, may need a joint approach to improve local organisation of care and how information is delivered across settings considering a "whole patient journey". Discharge summaries written by more senior staff in hospital highlighting potential red flags and greatest awareness in secondary care regarding the capabilities of primary care were suggested as elements that could improve the communication and transition between secondary and primary care. Commissioning and funding streams seem to be a major issue, as follow-up is recommended but not directly funded, unlike the pathways for cardiac and stroke rehabilitation, which were suggested by interviewees as models for post-ICU care. The evidence base for post-ICU follow-up is however partial and would benefit from further research. Interviewees highlighted that the specialised nature of post-ICU care meant that intensive care staff were better placed to understand and refer patient to services for cognitive, physical and mental health problems, but funding did not always allow this. Additionally, interviewees suggested services that were required for longer such as talking therapies and physical rehabilitation should be delivered in the community, where they might be more easily accessed.

VERSION 2 – REVIEW

REVIEWER	Vollam, Sarah University of Oxford, Nuffield Department of Neurosciences
REVIEW RETURNED	19-Apr-2021
GENERAL COMMENTS	Thank you for addressing my comments, I have nothing more to add.